# Characterizing Multimodal Long-form Summarization: A Case Study on Financial Reports

**Tianyu Cao**
Carnegie Mellon University
tianyuca@andrew.cmu.edu

**Natraj Raman & Danial Dervovic**
JPMorgan AI Research
{natraj.raman,danial.dervovic}@jpmorgan.com

**Chenhao Tan**
The University of Chicago
chenhao@uchicago.edu

## Abstract

As large language models (LLMs) expand the power of natural language processing to handle long inputs, rigorous and systematic analyses are necessary to understand their abilities and behavior. A salient application is summarization, due to its ubiquity and controversy (e.g., researchers have declared the death of summarization). In this paper, we use financial report summarization as a case study because financial reports are not only long but also use numbers and tables extensively. We propose a computational framework for characterizing multimodal long-form summarization and investigate the behavior of Claude 2.0/2.1, GPT-4/3.5, and Cohere. We find that GPT-3.5 and Cohere fail to perform this summarization task meaningfully. For Claude 2 and GPT-4, we analyze the extractiveness of the summary and identify a position bias in LLMs. This position bias disappears after shuffling the input for Claude, which suggests that Claude seems to recognize important information. We also conduct a comprehensive investigation on the use of numeric data in LLM-generated summaries and offer a taxonomy of numeric hallucination. We employ prompt engineering to improve GPT-4's use of numbers with limited success. Overall, our analyses highlight the strong capability of Claude 2 in handling long multimodal inputs compared to GPT-4. The generated summaries and evaluation code are available at `https://github.com/ChicagoHAI/characterizing-multimodal-long-form-summarization`.

## 1 Introduction

Summarization, the task of condensing the input text while preserving important information, is ubiquitous and has attracted a lot of interest in the AI community. Recent work demonstrates the strong capability of large language models (LLMs) in summarization. In fact, Pu et al. (2023) finds a clear preference for LLM-generated summaries over human-written ones and even declares the death of summarization.

Researchers have thus shifted to underexplored domains such as long-form summarization. For example, Chang et al. (2023) evaluates summaries of books and proposes novel measures of coherence. In these cases, human-generated summaries are rare, and it is no longer prudent to assume that human-generated summaries are golden standards. Furthermore, regardless of how good LLM-generated summaries are, it remains an open question *how* LLMs approach the summarization task.

In this work, we contribute to this recent line of work on evaluating long-form summarization by studying financial reports. Financial reports provide a great case study for two reasons. First, financial reports tend to be very long. In particular, the length of the most important section in 10-K reports, "Management's Discussion and Analysis of Financial Condition and Results of Operations" (MD&A), can already exceed the context window

of most current commercial models. Second, financial reports include many numbers and tables that are crucial for financial analysis, and this multimodal setting has been understudied in prior work. Thus, it is important to understand how large language models handle numbers in summarizing financial reports.

We propose a computational framework to characterize multimodal long-form summarization in summarizing financial reports. We analyze summaries from the state-of-the-art commercial models. As there is no gold standard summary, we are primarily interested in *how* LLMs summarize these very long inputs.

First, we investigate how the information in the reports is utilized. We quantify the extent to which the summaries are extractive and the source location for the extractive information. We find that extractive sentences represent 30% to 40% of the summary. Furthermore, these sentences tend to come from the beginning of the report, similar to the finding in Liu et al. (2023) about question answering, suggesting that LLMs have strong position biases. However, this position bias could be justified in our context because information at the beginning might be inherently more important. Indeed, the bias disappears for Claude after we shuffle the input, indicating that Claude seems to recognize important information in the input, whereas GPT-4 shows a consistent position bias towards the beginning of the input.

Second, given the important role of numbers, we conduct a thorough investigation on the use of numbers in LLM-generated summaries. We find that small models like GPT-3.5 tend to include generic texts without mentioning any specific numbers. In comparison, large models use numbers more extensively and often perform simple operations on numbers such as rounding and taking the difference. Claude 2 outperforms GPT-4 with a higher tabular numbers utilization ratio and a larger number density. We further provide a taxonomy of numeric hallucinations through manual annotation. The results show that although LLMs hallucinate about 5% of the time, they may fail to capture the semantic relationship between numeric data and textual data, an instance of *context mismatch*.

Finally, we explore the promise of prompt engineering to enhance the use of numbers by GPT-4. We find that GPT-4 extracts more numbers when prompted properly, but still uses fewer tabular numbers than the summary generated by a simple prompt with Claude 2. Also, fewer hallucinations are generated when using Chain-of-Thought prompting by GPT-4.

In summary, we make the following contributions:

- We develop a computational framework for characterizing multimodal long-form summarization that accounts for both information usage and numeric values.
- We demonstrate that Cohere and GPT-3.5 cannot perform such long-form multimodal summarization.
- We compare the behavior of Claude and GPT-4, and show that Claude demonstrates a stronger ability to use numbers and seems to recognize important information.
- We provide a taxonomy of numeric hallucination and investigate the promise of prompt engineering to tackle some of the weaknesses in these models.

## 2   Dataset and Methods

In this section, we introduce the dataset used in this work and provide details of how we generate summaries.

**Dataset.** Form 10-K filings serve as comprehensive annual reports that outline a company's financial condition, business overview, and other disclosures mandated by the U.S. Securities and Exchange Commission (SEC). An annual report on Form 10-K usually contains four main parts and sixteen specific items standardized by the SEC to ensure comprehensive coverage of key business aspects (SEC). Of particular interest for our analysis is Item 7, commonly referred to as "Management's Discussion and Analysis of Financial Condition and Results of Operations" (MD&A), where companies offer their perspectives on the preceding financial year's business outcomes.

| Model | R/S | Words | Num. Total | Dens.% | Num. A | Num. B | Num. C | Num. D |
|---|---|---|---|---|---|---|---|---|
| Claude 2.0 | R | 17146.70 | 478.78 | 2.79 | 136.90 | 295.75 | 46.14 | - |
| | S | 223.82 | 11.09 | 4.95 | 4.62 | 0.92 | 2.89 | 2.66 |
| Claude 2.1 | R | 17583.14 | 487.99 | 2.77 | 137.83 | 302.93 | 47.22 | - |
| | S | 229.66 | **11.56** | **5.03** | 4.72 | 0.97 | 3.11 | 2.76 |
| GPT-4 | R | 17439.68 | 485.50 | 2.78 | 137.63 | 301.08 | 46.79 | - |
| | S | **421.51** | 5.81 | 1.38 | 3.34 | 0.29 | 1.73 | 0.46 |
| GPT-3.5 | R | 4927.61 | 146.32 | 2.97 | 53.83 | 81.32 | 11.17 | - |
| | S | 120.12 | 0.46 | 0.38 | 0.28 | 0.03 | 0.14 | 0.01 |
| Cohere | R | 9893.39 | 260.74 | 2.64 | 98.28 | 139.68 | 22.79 | - |
| | S | 152.78 | 6.90 | 4.52 | 1.42 | 2.12 | 0.94 | 2.43 |

Table 1: Basic statistics of the reports and the summaries. "R" represents the report, and "S" represents the summary. "Words" denotes the average number of words in the report or summary. "Num." is the average number of numeric values. "Dens.%" represents the ratio of numbers over the number of words. "A", "B", "C", and "D" represent different categories of numeric data and are defined in §4.1.

We randomly selected 1,000 HTML files of 10-K forms obtained from the Electronic Data Gathering, Analysis, and Retrieval (EDGAR) system. These files are then converted into clean JSON format using EDGAR-CRAWLER (Loukas et al., 2021) and we use Item 7 as the target report to do summarization. Additionally, to explore whether LLMs have position biases, we generate shuffled versions of each report by randomly reordering its paragraphs, treating tables as individual paragraphs.

**Models.** We evaluate summaries generated by five commercial models with top context window length (wl) in July 2023 when we started this work: (1) **Claude 2.0** (`claude-2.0`, wl=100K tokens), (2) **Claude 2.1** (`claude-2.1`, wl=200K tokens),[1] (3) **GPT-3.5** (`gpt-3.5-turbo-1106`, wl=16K tokens), (4) **GPT-4** (`gpt-4-1106-preview`, wl=128K tokens) (Achiam et al., 2023), and (5) **Cohere** (`command`, wl=100K characters). For Claude 2.0 and Claude 2.1, we set max output tokens to 4,096 and temperature to 1. GPT-3.5 and GPT-4 are used with default hyperparameters. We use the Cohere `co.summarize` API and set the output summary length to "long", format as bullets, high extractiveness, and temperature to 0.3 according to the official recommendation. Except for Cohere, which does not need additional prompts with the specific endpoint, we use simple prompts for summarization. Detailed prompts are shown in Appendix D.

**General analysis of summary.** Table 1 presents basic statistics of the report and summary generated by each model. Notably, the summaries produced by GPT-4 exhibit the longest average word length of 421.51. With its substantial 200K-token context length, Claude 2.1 outperforms other models by extracting an average of 11.56 numeric values per summary. A more detailed investigation of numeric values utilization will be presented in Section 4. It is worth mentioning that certain reports require truncation to meet the context limit of different models, with an averaged of 436, 143, 12,656, and 7,690 words truncated for Claude 2.0, GPT-4, GPT-3.5, and Cohere respectively.

GPT-3.5 and Cohere are generally not good at summarization based on the simple prompt, reflected by too short summaries, meaningless contents, and referring to almost zero numbers. We investigate this further in Appendix A. Thus, the rest of this paper mainly focuses on Claude 2 and GPT-4.

## 3 Tracing the Summary in the Input

Obtaining ground-truth summaries for very long inputs is a significant challenge, and there can be numerous good summaries for a given document. Therefore, we mainly focus on the behavior of the models and propose a computational framework to characterize such

---

[1]We added Claude 2.1 when it was out, but Claude 3 came out in February 2024, so we have not been able to include it.

| Type | Sentences | Score |
|---|---|---|
| 1-1 | *summary sentence*: As of March 31, 2020, accumulated deficit was $112.3 million.
*report sentence*: As of March 31, 2020, we had an accumulated deficit of $112.3 million. | 1.70 |
|  | *summary sentence:* - For 2019, capital expenditures are estimated at $2.9 billion.
*report sentence:* We project our E&P capital and exploratory expenditures will be approximately $2.9 billion in 2019. | 0.95 |
| 2-1 | *summary sentence*: - Operating cash flows increased by $98 million due to higher net working capital inflows.
*report sentence 1*: The increase was primarily due to higher cash inflows for net working capital of $68.5 million and other current assets and liabilities of $28.2 million.
*report sentence 2*: Cash flows from operating activities increased $98.0 million in 2018 compared to 2017. | 0.94 |
|  | *summary sentence*:- Investing cash flows decreased due to lower cash paid for acquisitions.
*report sentence 1*: The decrease was primarily due to lower cash outflows for business acquisitions, net of cash acquired of $56.2 million, partially offset by higher cash outflows for capital expenditures of $30.1 million.
*report sentence 2*: Cash flows from investing activities decreased approximately $28.1 million in 2017. | 0.92 |
| Abstractive | *summary sentence*: Results of operations: A detailed comparison between the years 2018 and 2017 displays an increase in net income and net sales, while gross profit as a percentage of net sales experienced a decrease.
*report sentences*: Results of Operations - Consolidated\n Comparison of the years ended December 31, 2018 and 2017 \n For the year ended December 31, 2018, net income was $80.9 million, compared with net income of $57.8 million in 2017. Net sales increased by $215.7 million, or 15.4%, in the year ended December 31, 2018, compared with the prior year, with increased sales in Performance Coatings, Performance Colors and Glass and Color Solutions of $139.9 million, $42.8 million and $33.0 million, respectively. Gross profit increased $39.7 million, or 9.5%, in 2018 to $455.9 million, compared with $416.2 million in 2017 and, as a percentage of net sales, it decreased 150 basis points to 28.3%. | N/A |

Table 2: How generated summaries synthesize information from the input. Blue contents represent the matches from the first report sentence and green contents represent the matches from the second report sentence. "Abstractive" represents the abstractive sentences, where the report sentences were manually put together, and yellow is used for matches.

| Model

Pair | Original Report | | | Shuffled Report | | |
|---|---|---|---|---|---|---|
|  | Claude 2.0 | Claude 2.1 | GPT-4 | Claude-2.0 | Claude 2.1 | GPT-4 |
| 1-1 | 31.24% | 41.50% | 31.96% | 38.32% | 47.98% | 27.70% |
| 2-1 | 15.33% | 14.33% | 9.08% | 15.37% | 14.23% | 9.04% |

Table 3: Extractive summary sentences statistics. "1-1" represents 1-1 extractive summary sentence with one source report sentence and "2-1" for 2-1 synthesizing summary sentence with two source report sentences.

behavior. The first aspect is the extent to which the generated summaries are extractive. Second, we examine the information sources of the generated summaries, i.e., which part of the input a summary is derived from.

## 3.1 To What Extent Are Generated Summaries Extractive?

LLMs are generative models and thus in theory generate abstractive summaries. However, it is plausible that LLMs perform minimal paraphrasing of sentences in the input to produce the summary. Furthermore, extractive summaries can better maintain the fidelity to the original input. Therefore, we first attempt to match sentences in the summary to those in the input and measure to what extent generated summaries are extractive.

**Measuring extractiveness.** We modify the coverage measure in Grusky et al. (2018) to measure the similarity between two sentences with a greedy match algorithm. Given a summary sentence $S = \{t_{S1}, t_{S2}, ..., t_{Sn}\}$ and a report sentence $R = \{t_{R1}, t_{R2}, ..., t_{Rm}\}$ after removing stopwords, where $t_i$ is a token, the length of $S$ is $n$, and the length of $R$ is $m$.

At a high level, similarity$(S, R)$ is defined as the ratio of matched tokens with a quadratic bonus that rewards longer-matched sequences. Specifically, for each summary sentence, this algorithm processes its tokens iteratively, discovering and aligning the longest matching token sequences from each report sentence. Then, for each report sentence $R$, we get a list $M(S, R) = \{\{t_i, t_{i+1}, ..., t_{i+p}\}, ..., \{t_j, t_j, ..., t_{j+q}\}|t \in S \cap R\}$ consisted of the longest matching token sequences $m$ compared with the summary sentence $S$. The similarity score is computed as:

$$\text{similarity}(S, R) = \frac{1}{|S|} \sum_{m \in M(S,R)} (|m| + 0.1 * |m|^2) \ . \tag{1}$$

Following a meticulous manual check (Appendix F), we find sentences with a similarity score above 0.8 to contain almost verbatim excerpts from the source text, which we define as *extractive*. We calculate the similarity score between each summary sentence and report sentence. If the top similarity score is greater than 0.8, we consider the summary sentence a 1-1 extractive sentence. For each remaining summary sentence, we calculate a similarity score with a combination of two report sentences. If this score exceeds 0.8, we call it a 2-1 synthesizing sentence. Examples of 1-1 extractive sentences and 2-1 synthesizing sentences are shown in Table 2.

**Extractive sentences represent 30% to 40% of the summary.** Table 3 shows the percentage of extractive sentences. Claude 2.1 generates the most extractive contents with 41.50% of extractive summary sentences, which is much higher than 31.24% of Claude 2.0. This may be a result of the multi-level bullet formats of Claude 2.1. GPT-4 generates comparable extractive contents with Claude 2.0 but with a smaller percentage of 2-1 synthesizing sentences. Summaries generated by Claude 2 for the shuffled report exhibit more extractiveness, with 6-7% higher proportions of extractive sentences compared with that of original reports. In contrast, GPT-4 generates fewer extractive sentences from shuffled reports. This observation demonstrates the differing behavior of Claude and GPT-4 when processing incoherent texts.

**Analysis of the remaining abstractive sentences.** To gain insights into the abstractive capabilities of LLMs, we conduct a case study for the abstractive sentences present in the summaries. Specifically, we find GPT-4 generates abstractive summary sentences with a high degree of condensation, such as summarizing net income, net sales, and gross profit in a single sentence (Table 2). It paraphrases the original information across various locations and compacts it into one single sentence, generating new phrases such as "display an increase in net income". The entire meaning is retained, but the content organization is significantly different from the original report. See the Appendix G for similar examples from Claude 2.

## 3.2 Where Does the Information Come From?

We study this question by examining the distribution of source contents for the summary.

**Identify source contents.** Based on the extractive analysis, we analyze the distribution of the 1-1 extractive sentences. For each summary sentence, the report sentence with the highest similarity score is treated as its source. If there exist several report sentences with the same score, we choose the one with the top cosine-similarity using sentence embeddings generated by SentenceTransformers (Reimers & Gurevych, 2019). Then we calculate the position of each source report sentence and visualize the distribution in Figure 1. We also run the same analysis by including the location of 2-1 synthesizing summary sentence sources. The results are similar and presented in Appendix B.

**Most summary information comes from the beginning.** Figure 1(a) shows that a high percentage of source contents are found at the first 20% of the report——models tend to use contents at the very beginning of the input, and pay less attention to the information within the middle and the end part of its context. For example, more than 60% of the summary contents generated by GPT-4 come from the first 20% of the report, and less than 10% of the summary contents are from the middle. GPT-4 also shows a preference for the information at the very end of the report, which is not as salient in the summary generated by Claude 2.

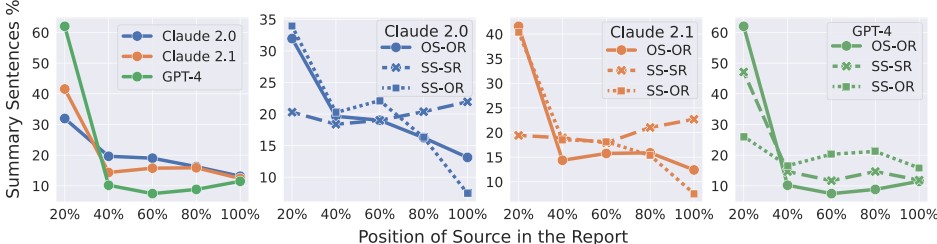

(a) Source information
distribution of different models

(b) Comparison of source information distribution
before & after shuffling, and reordered

Figure 1: Source distribution of extractive summary sentences. OR and SR stand for original report and shuffled report respectively, while OS and SS stand for summaries of the original report and shuffled report respectively. For the summary generated from the original report, most information comes from the beginning of the report. However, for the shuffled reports, this trend disappears for Claude but stays for GPT-4.

**Shuffled report analysis.** To understand whether the observed pattern is a bias in LLM's behavior or a consequence of document structure (there is an overview at the beginning of Item 7), we analyze the summaries after we shuffle the input reports. Figure 1(b) SS-SR shows that the source information is evenly distributed for Claude 2. The distribution SS-OR becomes heavy in the beginning when comparing the summary of the shuffled report with the original report, suggesting that Claude 2 seems to recognize the important information even after shuffling. In comparison, GPT-4 maintains a strong position bias for the shuffled input: the summary of the shuffled input now favors the beginning part despite that it is essentially arbitrary text from the input.[2]

In summary, although we observe a strong position bias, it is not "lost in the middle" as in Liu et al. (2023). In fact, our analysis reveals intriguing behavioral differences between Claude 2 and GPT-4.

# 4 Numeric Values Utilization

Multimodal financial reports convey crucial information through a combination of unstructured textual narratives and structured tabular data. It is important to effectively integrate both data modalities in the summaries. Meanwhile, it is commonly suggested that LLMs can have issues with using numbers (Dziri et al., 2024). Thus, we present a detailed analysis of how LLMs leverage numeric values from both texts and tables in financial reports.

## 4.1 Basic Numeric Values Analysis

To extract numeric values from the reports, we use a regular expression that matches numbers with commas as thousand separators and an optional decimal part. We excluded numbers from entity names (e.g., "COVID-19", "ATA190"), dates (e.g., "December 31, 2022"), and residual HTML table indices, focusing primarily on financially meaningful values (see Appendix C for details).

Based on the source of the numeric values appearing in the summaries, we categorized them into four types:

- Type A: Numeric values present only in the report's text.
- Type B: Numeric values present only in the report's tables.
- Type C: Numeric values present in both the text and tables of the report.
- Type D: Numeric values not found in the report.

One may hypothesize that it is challenging for LLMs to incorporate information from the table. Thus, numeric values of type A may be most likely to show up in the summaries,

---

[2]Due to cost reasons, the GPT-4 analysis is based on 100 shuffled reports.

while numbers in type B are rarely incorporated. Alternatively, the model might consider type C to be the most important.

**Claude 2 demonstrates a more sophisticated use of numbers than GPT-4.** As illustrated in Table 1, Claude 2.1 demonstrates a strong ability to use tabular numeric values, with 8.37% of its numeric values belonging to type B, compared to only 4.98% for GPT-4. The difference is statistically significant according to a two-sample $t$-test ($p < 0.001$). However, it is noteworthy that in the reports, more than half of the numeric values were exclusively present in tabular formats, suggesting that LLMs tend to prioritize numeric values found in the text and pay relatively less attention to tabular data.

Similarly, when comparing the overall density of numeric values in the summaries, Claude 2.1 uses numbers more frequently than GPT-4, with a summary number density of 5.03% versus 1.52% ($p < 0.001$). This discrepancy can be attributed to GPT-4's tendency to generate longer summaries with fewer numeric values, indicating a potential weakness in analyzing and incorporating numeric information, especially from tabular data, compared to Claude.

As we mentioned earlier, GPT-3.5 and Command fail to use numbers in the summaries. Specifically, GPT-3.5's summaries do not include numeric values in 831 of 1,000 reports and only an average of 0.46 number is mentioned. For 46.7% of the reports, Cohere simply generates "unable to comprehend the table". However, for those completed summaries, it extracts an average of 6.90 numbers, which is larger than that of GPT-4. Among those, Command prefers numbers from the tables rather than text (2.12 vs. 1.42), which is quite different from the behavior of other models.

**Type D numbers can come from numeric operations.** Numeric values of type D, which represent mismatches between the summary and the report, accounted for an average of 2.76 instances per summary generated by Claude 2.1 and 2.66 for Claude 2.0. In contrast, GPT-4 has only 0.46 type D numbers. Through human annotation, we identified that these mismatches often stem from simple operations such as rounding, calculating differences, or computing rates of change (e.g., 86.36% of type D numbers generated by Claude 2.0 come from such operations).

One notable example that showcases the LLMs' ability to synthesize numeric information is their accurate representation of numeric values in the correct units, such as billions instead of millions, as shown in Figure 2. This exemplifies the LLMs' ability to extract and retain crucial details, coupled with their proficiency in comprehending and manipulating numeric data coherently throughout the summarization process. However, it is important to note that a subset of these type D numeric values could potentially be instances of numeric hallucinations, a phenomenon that warrants further investigation, as discussed in Section 4.2.

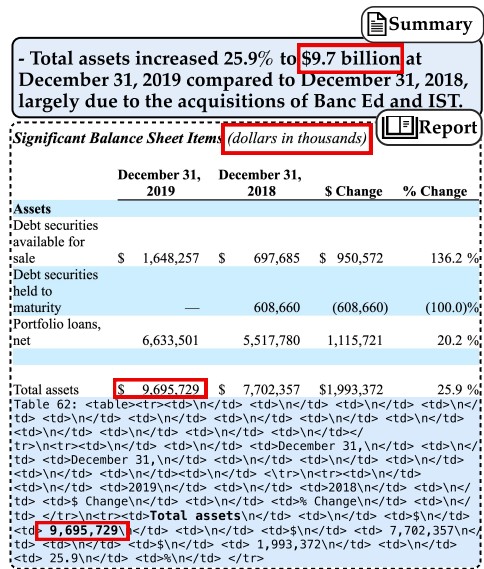

Figure 2: The summary sentence presents a rounded figure with the unit adjusted as per the remark preceding the table. The input tables are in HTML format as shown in the figure's bottom section.

## 4.2 Numeric Hallucinations

Numeric hallucinations refer to cases where a numeric value in the generated output text is inconsistent with or unsubstantiated by the corresponding source data. Although numeric values of type D are reasonable candidates for hallucinations, we find that other numbers

could be hallucinations too, e.g., referring to 75 million in profit as 75 million in debt. Therefore, we resort to manual annotation to assess the type and extent of hallunication.[3]

**Annotation protocol.** To comprehensively annotate numeric hallucinations in summaries of lengthy financial reports, we employ the following multi-step protocol:

(1) Candidate extraction: For each numeric value mentioned in the summary, extract all sentences (quotes) from the source report that contain either the exact same numeric value or values in different formats (e.g., 1,000,000 and 1M).
(2) Match identification: The annotator examines the list of extracted report quotes for a given summary number:
   a. If a quote accurately matches the context and usage of the numeric value in the summary, it is annotated as "No Hallucination".
   b. If no matching quote is found, the annotator conducts a comprehensive search through the entire report.
   b1. If a correct match is identified, the corresponding quote is recorded, and the instance is annotated as "No Hallucination."
   b2. If no substantiating evidence is found in the report, the instance is annotated as a specific type of numeric hallucination based on the taxonomy definitions.
(3) Annotations and comments: For cases annotated as a hallucination type, the annotator provides detailed comments explaining the rationale behind the annotation decision.

This protocol ensures a systematic and thorough annotation process, aiding in the accurate identification and categorization of numeric hallucinations. It emphasizes examining all potential evidence from the source report before finalizing annotations. The lead author performed this analysis and annotated numbers in the summaries of a random sample of 20 reports for Claude 2.0, Claude 2.1, and GPT-4.

**A taxonomy of numeric hallucination.** We identify and categorize four types of numeric hallucinations, collectively forming the first taxonomy for this phenomenon:

- Fabricated Number: The presence of a specific numeric value in a generated text that lacks any corresponding references.
- Rounding Error: Discrepancy arising from the rounding off of numeric values.
- Arithmetic Error: Numbers generated from incorrect mathematical calculations applied to numeric values from the report.
- Context mismatch: An inconsistency where the same numeric value applies to different contexts in the report vs. in the summary.

Instances of each type of hallucination are shown in Table 4.

**LLMs hallucinate in only about 5% of numerical values.** Overall, LLMs hallucinate at a low percentage when using numeric values, with total hallucination rates of 6.48%, 3.97%, and 5.74% observed for Claude 2.0, Claude 2.1, and GPT-4, respectively. Context mismatch hallucinations exhibited the highest percentages across all models, which suggests that current models still face challenges in accurately capturing the semantic relationships between numeric data and their corresponding textual descriptions when generating summaries. In contrast, fabricated number hallucinations occurred at relatively low percentages, indicating that most models demonstrated reasonable performance in avoiding blatantly unsupported numeric claims in their outputs. However, even low rates of such hallucinations can undermine the trustworthiness of the summaries.

**Performance analysis: GPT-4 vs. Claude 2.** As shown in Table 4, GPT-4 showcased impressive performance by generating zero instances of fabricated numbers and arithmetic errors. This can be attributed to GPT-4's tendency to produce extractive summaries, where most numeric values are directly copied from the source report, as evidenced by the numeric analysis in Section 4.1. In contrast, Claude 2 attempted to generate more abstractive content, involving a higher degree of arithmetic operations, which contributed to its presence of arithmetic error hallucinations. Notably, Claude 2 exhibited a lower percentage of

---

[3]We experimented with automatic analyses using GPT-4, but GPT-4 has trouble making sense of these statements accurately.

| Type | Example | % |
|---|---|---|
| Fabricated number | *Summary:* - Net income for 2018 was $39 million, compared to $33 million in 2017.
*Report:* No exact source. | 0.81/0.79/0 |
| Arithmetic error | *Summary:* - the company has $750 million in revolving credit facilities and $600 million in unsecured term loans to fund operations and acquisitions.
*Report:* Our principal debt obligations at December 31, 2015 consisted of borrowings under our $750,000 unsecured revolving credit facility, our $300,000 unsecured term loan, our $250,000 unsecured term loan, $350,000 of publicly issued senior unsecured notes and five secured mortgage loans that were assumed in connection with certain of our acquisitions.
*Comments:* 300,000 + 250,000 = 550,000 | 1.21/0.40/0 |
| Rounding error | *Summary:* - net loss was $234 million compared to net loss of $87 million in 2014, driven by $191 million impairment charge related to the speaker & receiver product line and $144 million impairment of intangible assets.
*Report:* `<td>Impairment of intangible assets\n</td> <td>\n</td> <td>144.7\n</td>` | 0.40/1.19/1.64 |
| Context mismatch | *Summary:* - marketplace subscription revenue represented 89% of 2018 revenue.
*Report:* Marketplace subscription revenue increased $123.1 million in the year ended December 31, 2018 compared to the year ended December 31, 2017, and represented 89% of total revenue in both 2018 and 2017. | 2.43/1.59/3.28 |

Table 4: Taxonomy of numeric hallucinations and examples for each. The last column % represents the hallucinated numbers percentage of all annotated numbers from summaries generated by Claude 2.0, Claude 2.1, and GPT-4. Red represents the hallucinated number and blue represents the correct source.

context mismatch hallucinations compared to GPT-4, suggesting a more consistent semantic grounding when quoting numeric values from the report within the generated summaries.

## 5 Prompt Engineering to Improve Use of Numbers

To enhance GPT-4's performance in extracting numeric values, particularly from tables, we design three explicit prompts and one chain-of-thought (CoT) prompt (Appendix D): **NUM** to explicitly request the inclusion of numeric values. **TAB** to explicitly request the inclusion of tabular numbers. **NUMTAB** to explicitly request both numeric values and tabular numbers. **CoT** to give intermediate reasoning steps of using numeric values.

As shown in Table 5, the NUMTAB prompting strategy enables GPT-4 to extract the highest number of numeric values and achieve the highest summary number density of 4.48% among the five prompt strategies. However, it still falls short of the 5.03% density achieved by Claude 2.1 using a simple prompt. Notably, the CoT prompting substantially enhances GPT-4 using tabular numbers, with 19.47% of the summary numbers being extracted exclusively from the report tables.

To assess the effectiveness of CoT prompting in reducing hallucinations, we annotated a random sample of 10

| Model | Prompt | Total | Dens.% | Percentage % | | | |
|---|---|---|---|---|---|---|---|
| | | | | A | B | C | D |
| GPT-4 | simple | 5.81 | 1.38 | 57.47 | 4.91 | **29.77** | 7.85 |
| | NUM | 14.30 | 3.38 | 56.64 | 4.90 | 24.48 | 13.99 |
| | TAB | 9.40 | 2.19 | **62.77** | 2.13 | 25.53 | 9.57 |
| | NUMTAB | **18.90** | 4.48 | 55.03 | 6.88 | 21.16 | 16.93 |
| | CoT | 11.30 | 3.49 | 43.36 | **19.47** | 28.32 | 8.85 |
| Claude 2.1 | simple | 11.56 | **5.03** | 40.81 | 8.37 | 26.91 | **23.88** |

Table 5: Numeric statistics for the summary generated by different prompts.

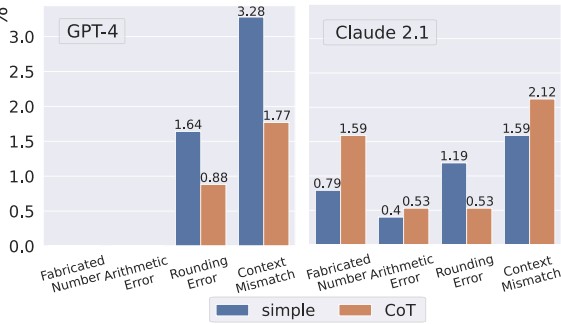

Figure 3: Hallucinated numbers frequency of summaries generated by simple prompt and CoT prompt respectively.

summaries generated by both GPT-4 and Claude 2.1. As shown in Figure 3, much fewer hallucinations are generated when using CoT prompts to guide GPT-4 on avoiding specific

numeric hallucinations, but this is not the case for Claude 2.1. This shows GPT-4's better ability to leverage CoT prompting for introspecting its own output.

## 6 Related Work

Evaluation of long-form summarization is very challenging as traditional evaluation requires reference summaries (Lin, 2004; Papineni et al., 2002). The closest work to ours is Chang et al. (2023), which presents the first study of LLM-based book-length summary. Their focus is on coherence, while our work focuses on the characterization of LLM behavior in a multimodal setting. Other work has shown that summaries generated by GPT-3 and current LLMs are overwhelmingly preferred by humans and these also do not suffer from common dataset-specific issues such as poor factuality (Goyal et al., 2023; Pu et al., 2023).

A battery of studies examine automatic evaluation using LLMs considering the high cost of human annotation (Min et al., 2023; Peng et al., 2023; Gilardi et al., 2023, *i.a.*). However, Doostmohammadi et al. (2024) suggest that automatic evaluation can only approximate human judgments under specific conditions, and their reliability is highly context-dependent.

## 7 Conclusion and Discussion

We present the first study to characterize summaries generated by LLMs for multimodal long-form inputs. Our results reveal that different LLMs may approach this task quite differently in terms of extractiveness, position bias, and use of numbers. While the position bias of GPT-4 is probably sub-optimal, it is unclear what makes a good summary, especially for such long-form inputs. On the common concern about hallucinations, we find that hallucinations do not happen very frequently. Future work may not only focus on the frequency but also the potential impact of such misuse. The underlying causes of our observations also remain unclear; we hypothesize that Claude may have more exposure to enterprise data/applications involving structured data and business metrics. This exposure could potentially explain Claude's stronger ability to use numbers in the summary. However, it is important to note that all the LLMs used in our work are closed and the hypotheses need further investigation.

Our research community urgently need novel perspectives for evaluating summarization. The "death" of summarization is an opportunity for advancing summarization beyond simple relevance and human preference.

## Acknowledgements

We thank insightful comments from anonymous reviewers and members of the Chicago Human+AI Lab, especially Chao-Chun Hsu for his early contribution in building the dataset. We sincerely thank Kaiqiao Han, Haoran Deng, and Yang Yang for their valuable advice on our work. This work has been supported in part by J.P. Morgan AI Faculty Research Award. This paper was prepared for informational purposes in part by the Artificial Intelligence Research group of JPMorgan Chase & Co and its affiliates ("J.P. Morgan") and is not a product of the Research Department of J.P. Morgan. J.P. Morgan makes no representation and warranty whatsoever and disclaims all liability, for the completeness, accuracy or reliability of the information contained herein. This document is not intended as investment research or investment advice, or a recommendation, offer or solicitation for the purchase or sale of any security, financial instrument, financial product or service, or to be used in any way for evaluating the merits of participating in any transaction, and shall not constitute a solicitation under any jurisdiction or to any person, if such solicitation under such jurisdiction or to such person would be unlawful.

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

# A  Analysis of GPT-3.5 and Cohere's Performance

| Model | R/S | avg | std | min | max |
|---|---|---|---|---|---|
| **GPT-3.5** | report | 4927.61 | 139.03 | 3465 | 5000 |
|  | summary | 120.12 | 39.35 | 26 | 358 |
| **Cohere** | report | 9893.39 | 179.64 | 8850 | 10000 |
|  | summary | 152.78 | 30.32 | 13 | 231 |

Table 6: Summary length statistics of GPT-3.5 and Cohere. All the numbers represent the number of words.

## A.1  Cohere

Among all the 1000 summaries, 46.7% only generate "I'm sorry, but I am unable to complete your request because I am unable to identify the relevant information to complete the task." This suggests that Cohere fails to deal with tasks in which the input length is almost equal to the context limit. However, to be noticed, Cohere is quite good at reading tables in its completed summary, where 30.69% of the numbers are extracted only from tables.

## A.2  GPT-3.5

As shown in Figure 6, due to the context limit of GPT-3.5, it could only accept reports with an average length of 4827.61 words and all the reports have to be truncated before summarization. On average, GPT-3.5 generates summaries that are only 120.12 words long due to its shorter input compared to other models. What's more, 831 summaries contain zero numbers and each summary only uses 0.03 tabular numbers on average, which demonstrates that GPT-3.5 fails to process multimodal documents.

# B  Source Information Distribution of 2-1 Synthesizing Summary Sentences

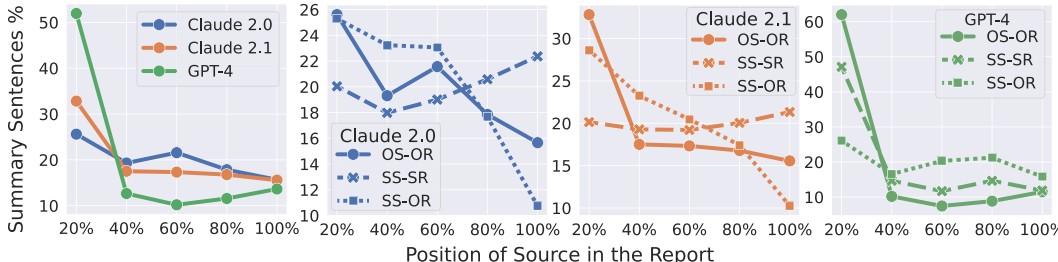

Figure 4: Source location distribution of 2-1 synthesizing sentences.

# C  Numeric Values Extraction Details

Dates number:

```
r'(January|February|March|April|May|June|July|August|September|October|
    November|December) \d{1,2},'
```

HTML table indices:

```
r'Table \d+:'
```

Target numeric values:

- Integer numbers with or without commas as thousands separators.
- Decimal numbers with or without commas as thousands separators and with or without a fractional part.

```
r‘(?<!\d)(?<![a-zA-Z-])\d{1,3}(?![a-jln-zA-JLN-Z\d])(?:,\d{3})*(?:\.\d+)?’
```

When extracting numeric values from the report or summary, numbers from entity names (e.g., "COVID-19", "ATA190"), dates number (e.g., "December 31, 2022"), and residual HTML table indices are excluded, focusing primarily on financially meaningful values.

## D  Prompts

**Simple prompt**

```
Summarize the following report.

MD&A:

---
The following is an MD&A report:

...

Please summarize this report.
```

**NUM prompt**

```
Summarize the following report.

MD&A:
...

Please include specific numeric values and key statistics.
```

**TAB prompt**

```
Summarize the following report.

MD&A:
...

Please include the numeric values in the tables.
```

**NUMTAB prompt**

```
Summarize the following report.

MD&A:
...

Please include specific numeric values and key statistics. Please include
the numeric values in the tables.
```

**Chain-of-Thought (CoT) prompt**

```
Summarize the following report.
```

```
MD&A:
...

Let's generate the summary step by step.

1. Read through the entire MD&A report carefully to understand the context.
2. Identify and extract the key topics and insights discussed in the
report.
3. Pay attention to any tables presenting numeric data, such as income
statements, balance sheets, or cash flow statements.
4. When including numbers in the summary, ensure they are:
    a) Explicitly stated values from the original report (do not fabricate
numbers).
    b) Stemmed from step-by-step verified calculations.
    c) Correctly rounded.
    d) Appropriately represented with clear context from the original
source.
5. Synthesize the extracted information and numbers into a concise summary
that flows logically.

Summary:
```

# E   Details of Similarity Score Calculating Algorithm

```
function greedy_match(summary_sentence, report_txt):
    S ← [], F ← {}

    // Preprocessing
    for each word w in tokenize(summary_sentence):
        if w ∉ stop_words ∧ w ∉ punctuation:
            if w contains both letters and digits:
                (w_digits, w_letter) ← w
                Add w_digits, w_letter to S
            else:
                Add w to S

    // Matching
    for each report sentence r_sen in report_txt:
        i ← 0
        while i < length(S):
            R ← [w for w in tokenize(r_sen) if w ∉ stop_words ∧ w ∉ punctuation]
            j ← 0
            seq ← []
            while j < length(R):
                if S[i] = R[j]:
                    i_prime ← i
                    j_prime ← j
                    while i_prime < length(S) ∧ j_prime < length(R) ∧ S[i_prime] = R[j_prime]:
                        i_prime++
                        j_prime++
                    if length(seq) < i_prime − i:
                        seq ← S[i, i_prime]
                    j ← j_prime
                else:
                    j++
            if length(seq) ≠ 0:
                if r_sen ∈ F:
                    Append seq to F[r_sen]
                else:
                    Add (r_sen, [seq]) to F
            i ← i + max(length(seq), 1)
    s_token_len ← length(S)
    r_sen_list ← []

    for r_sen, token_seq_list in F.items():
        matched_seq_len ← 0
        for token_seq in token_seq_list:
            matched_seq_len += (length(token_seq) + 0.1 * length(token_seq) * length(token_seq))
        similarity_score ← matched_seq_len / s_token_len
        Add (r_sen, similarity_score) to r_sen_list
    r_sen_list ← sorted(r_sen_list)
    top_r_sen, max_similarity_score ← r_sen_list[0]

    // Use SentenceBERT embedding to choose the top match report sentence with same similarity score
    summary_embedding ← get_sentBERT_embedding(summary_sentence)

    for r_sen, similarity_score in r_sen_list:
        if similarity_score == max_similarity_score:
            top1_embedding ← get_sentBERT_embedding(top_r_sen)
            sen_embeddding ← get_sentBERT_embedding(r_sen)
            cos1 ← cosine_similarity(summary_embedding, top1_embedding)
            cos2 ← cosine_similarity(summary_embedding, sen_embeddding)

            if cos2 > cos1:
                top_r_sen ← r_sen
```

Figure 5: Greedy match algorithm for similarity score calculation.

## F  Sample Sentence Pairs of Different Similarity Scores.

| score range | sentence pairs | score |
|---|---|---|
| 0.6-0.7 | *summary sentence:* Boat segment sales declined due to production disruptions from covid-19, but rebounded in the second half of 2020. 
 *report sentence:* The boat segment operating earnings were $70.2 million in 2020, a decrease of 7.9 percent compared with 2019, due to lower net sales along with unfavorable impact of absorption resulting from production disruptions, which were partially offset by benefits from cost reduction measures. | 0.67 |
| | *summary sentence:* - Free cash flow was $629 million in 2020, enabling debt paydown, share repurchases and dividends. 
 *report sentence:* Generated free cash flow of $629.3 million in 2020 enabling us to execute our capital strategy as follows: | 0.65 |
| 0.7-0.8 | *summary sentence:* - Total operating expenses increased 2.0% to $42.7 billion, driven by higher salaries, maintenance, and regional capacity costs. 
 *report sentence:* The year-over-year increase in our pre-tax income on both a gaap basis and excluding pre-tax net special items was principally driven by higher revenues and lower fuel costs, offset in part by increases in salaries, wages and benefits, maintenance expenses and costs associated with increased regional capacity. | 0.74 |
| | *summary sentence:* - At December 31, 2016, nhi had $2.6 billion invested in 205 facilities across 32 states. 
 *report sentence:* At December 31, 2016, we had investments in real estate, mortgage and other notes receivable involving 205 facilities located in 32 states. | 0.73 |
| 0.8-0.9 | *summary sentence:* - American reached a confidential settlement with Boeing regarding financial damages from the 737 max grounding. 
 *report sentence:* As previously announced in January 2020, we reached a confidential agreement with Boeing on compensation related to financial damages incurred in 2019 due to the grounding of the Boeing 737 max aircraft. | 0.85 |
| | *summary sentence:* - Cash from operations dropped to $209 million in 2019 from $389 million in 2018 due to lower earnings. 
 *report sentence:* Cash flows from operating activities decreased to $209.1 million in 2019 compared to $389.0 million in 2018 primarily due to lower earnings, partially offset by a favorable changes in working capital. | 0.82 |
| 0.9-1.0 | *summary sentence:* Passenger revenue was $42.0 billion, up 3.3% due to higher passenger demand. 
 *report sentence:* Passenger revenue increased $1.3 billion, or 3.3%, in 2019 from 2018 due to continued strength in passenger demand resulting in an increase in RPMs and a year-over-year increase in passenger load factor. | 0.92 |
| | *summary sentence:* - Company restaurant revenues come from food and beverage sales at company-operated restaurants. 
 *report sentence:* Company restaurant revenues consist of sales of food and beverages in company-operated restaurants. | 0.94 |
| >1.0 | *summary sentence:* Net earnings were $374.7 million in 2020 versus $30.4 million in 2019. 
 *report sentence:* Net earnings from continuing operations increased to $374.7 million in 2020 from $30.4 million in 2019. | 1.13 |
| | *summary sentence:* Franchise royalty and other franchise revenues come from royalty income and initial and renewal franchise fees. 
 *report sentence:* Franchise royalty and other franchise revenues represents royalty income, and initial and renewal franchise fees. | 1.38 |

Table 7: Sample sentence pairs of different similarity scores.

## G More Samples of Abstractive Sentences

| Abstractive Sentence |
| --- |
| *summary sentence*: The decrease was primarily due to lower sales in the industrial solutions and fluid handling solutions segments, partially offset by higher sales in energy solutions.
*report sentences*: The decrease is primarily attributable to decreases of $24.5 million in our Industrial Solutions air pollution control technologies, $18.2 million in custom-designed FCC cyclone systems serving the refinery markets, and $4.4 million in volume decreases in the Company's filtration and pump solutions sales. These decreases in net sales are offset by increases of $10.1 million in the Company's custom acoustical technologies that serve the natural gas power generation markets, $8.1 million in volatile organic compounds ("VOC") abatement solutions from the EIS acquisition and $3.3 million in volume increases in our emissions management and water filtration solutions technologies. |
| *summary sentence*: - The company has a credit facility with $72.6 million drawn and $104.7 million of available capacity as of December 31, 2020.
*report sentences:* `<td> Total outstanding borrowings under Credit Facility\n</td> <td>\n</td> <td>\n</td> <td> 72,616\n</td> <td>\n</td> <td>\n</td> <td>\n</td> <td> 65,501\n</td> <td>\n</td> </tr>\n<tr>`
In 2020, the Company made repayments of $2.5 million on the term loan and net borrowings on the revolving credit lines of $9.5 million, consisting of $10.3 million used to fund the EIS acquisition on June 4, 2020 and $2.6 million used to pay term debt assumed in connection with the Mader joint venture, offset by $3.4 million in net repayments on the revolving credit facility. |

Table 8: Samples of abstractive sentences.

