# OpenReview forum: "Characterizing Multimodal Long-form Summarization: A Case Study on Financial Reports"
_colmweb.org/COLM/2024/Conference — COLM_

### Official Review · Reviewer_KNTY · 2024-05-10

**Rating:** 7
**Confidence:** 3
**Ethics Flag:** 1

**Summary:**

The paper presents a case study on summarization capabilities of LLMs. Specifically, the authors focus on summarization of financial reports with several current top commercial LLMs (GPT, Claude, Command).

The study focuses a lot on processing numbers, which are known to pose various difficulties to LLMs. The authors perform various analyses, including a small-scale manual annotation of number hallucinations. In general, the authors find that number processing is very strong in some of the models, especially Claude.

The study also analyzes positional bias with LLMs tending to mostly cover information from the beginning of the document, and find interesting differences between Claude and GPT.

**Questions To Authors:**

These are something between questions and tentative suggestions (which might or might not be good suggestions, so this is just for consideration).
- Is text+tabular data generally called "multimodal" data? I can understand that, but I am more accustomed to "multimodal" referring to other modalities than only text, while tables are still text. But maybe my terminology is wrong and yours is correct, I have not specifically focused on working with tabular data so my terminology is not strong there.
- In table 1, would it make sense to report A B C D as percentage instead of absolute numbers, probably a sort of recall of what proportion of the numbers from the report appear in the summary?
- Also, table 1 is quite far from it description, could it maybe be moved to a lter part of the paper?
- Also, A B C D are not very reader-friendly numbers, could they be maybe e.g. X B XB N (teXt taBle teXt+taBle None) or something more interpretable?

**Reasons To Accept:**

- The study presents useful findings about the behavior of LLMs, partially in line with previous research, partially providing different or more complex observations than previous research, and partially adding new observations that had not been previously reported.
- The manual analysis of number hallucinations is particularly interesting, delving quite deep and showing various types of number hallucinations (with some not actually being hallucinations but rather new numbers correctly derived by the LLM from the text).
- The observed differences in the position bias are new and quite intriguing. This gives us a new finding that probably GPT4 is not great at summarizing documents where the important information is not at the beginning but elsewhere.
- Although the study focuses on one specific document type, I believe the findings are probably generalizable to a great extent and are thus of general interest and usefulness.
- The findings are not interpreted in detail, which provides nice future work avenues for trying to explain the observations presented.

**Reasons To Reject:**

- The study is only descriptive and observational, it covers in great detail the observed behavior and capabilities of the models, but does not attempt at all at interpreting the findings or hypothesizing about the reasons that lead to the findings. Of course this is somewhat hard since all of the investigated systems are closed-source with only limited information available about them, but still some information is provided by the models' authors and other information have been reported by independent researchers, so it would be good to at least try to link the known differences in the models' design and training to the observed differences in behavior on the examined task.
- The study mostly does not attempt to evaluate the quality of the reports, claiming there is no gold standard summary for their inputs. This is sad, as although we observe some differences in the models, in many cases we do not know for sure whether the differences mean that one of the models produces better summaries than the other, or whether these are just two possible approaches to the summarization task which are both good. I understand the difficulty in not having the gold standard data, but this should at least be discussed more explicitly and in more detail. For example, the fact that a model produces less numbers in the report, does this automatically mean the report is worse, or may it be that it is a highly abstractive report that interprets the numbers textually without mentioning them and thus this might also be a great report? I am not knowledgeable in the domain of financial reports and their summaries, so I do not know whether a good summary absolutely must contain a lot of numbers (which is somehow indirectly implied by the paper but not explicitly said or discussed). So this is a clear limitation of the paper. Still, I appreciate the manual evaluation of number hallucinations, which goes beyond this and actually does evaluate what is good and what is bad.
- For models with a short input context window, the authors simply truncate the input. This does not seem right, why not use a sliding window approach?

---

> ### Author Rebuttal · Authors · 2024-05-29
>
> Thank you very much for the insightful comments! We appreciate your recognition of our study on position bias and numeric hallucinations as well as the potential generalizability of our results.
>
> **Re: Hypotheses for the findings**
>
> Thank you for your insightful suggestions. We provide the following hypotheses and will add to the revision.
>
> - Claude may have more exposure to enterprise data/applications involving structured data and business metrics. The objectives during the fine-tuning of Claude might include tasks that emphasize utilizing numerical values. Therefore, it may employ numbers in a more sophisticated manner than GPT-4.
> - Most documents follow a pattern of introducing key points early and GPT-4 may have followed it. Claude's training/fine-tuning data may have incentivized extracting crucial information as summaries. It may also develop context-based summarization strategies rather than over-relying on position.
>
> **Re: Evaluation of the quality of the reports.**
>
> Summary quality is often dependent on the target application/task, hence we do not provide quality scores, which may be misleading. Our study focuses on proposing a computational framework for characterizing multimodal long-form summarization. By investigating *how* LLMs approach the summarization task, we believe that our work is a meaningful first step towards this goal.
>
> **Re:  About sliding window approach**
>
> We agree that using a sliding window to do long-form summarization is a great idea. However, because the best models can handle most of the input (200K tokens window length for Claude 2.1) and these models boast their ability to handle long inputs, so we would like to evaluate that directly.
>
> **Re: Questions**
>
> - Tabular data is structured while text is unstructured. And they should be processed differently when doing summarization. Therefore, Text+table data is treated as multimodal data.
> - Using exact quantitative values highlights the large differences in number utilization across models. For instance, while Claude 2.1 cites 0.97 tabular data, GPT-4 only cites 0.03. Using percentages alone may not capture as clearly.
> - Thank you for the insightful suggestion. Table 1's model, summary length, and total numbers information help readers learn about our work generally at the very first of the paper. We will try to improve readability by moving the numeric statistics data to a later section.
> - This is a great idea and we will modify it in our final version.

---

> > ### Comment · Area_Chair_dLMQ · 2024-06-03
> > **[Area Chair comment] Author response**
> >
> > Dear reviewer ,
> >
> > Please take a look at the author's rebuttal. If the rebuttal addressed your concerns, please let the authors know about this and update your review. If not, please continue to engage with the authors and the other reviewers in the discussion forum.
> >
> > Are your concerns about the descriptive vs prescriptive nature of the results addressed? (Reviewer Lb8C also had similar issues).
> >
> > Other reviewers should also feel free to comment on these points!
> >
> > Thanks.

---

> > ### Comment · Reviewer_KNTY · 2024-06-06
> >
> > Thanks a lot for the reply, which has addressed all of my major comments. I quite like the replies, and I think the paper can be made yet a little better by reflecting some of those in the final version of the text.
> >
> > "Re: Hypotheses for the findings" -- thanks a lot for that, that's what I was looking for. Although I see now that this is necessarily quite vague, so I understand why you did not put this in the original paper, and I leave it up to your decision what to include and what is too vague to include. (Myself I would add the hypothesis about Claude's possible "exposure to enterprise data/applications involving structured data and business metrics" which I find insightful and sensible, but the rest maybe is too vague.)
> >
> > I already gave the paper the "Good paper, accept" rating, and I stand by this rating.

---

> > > ### Author Response · Authors · 2024-06-06
> > >
> > > Thank you for the positive feedback and we will incorporate the hypothesis in the final version of the paper!

---

### Official Review · Reviewer_Lb8C · 2024-05-10

**Rating:** 6
**Confidence:** 4
**Ethics Flag:** 1

**Summary:**

The paper conducts a comparison of various LLM models regarding their ability to summarize sections of financial reports (10-K), which encompass not only textual content but also tabular and numeric data. It highlights two key findings. Firstly, it identifies a position bias among all models, indicating a tendency to prioritize the initial sentences. Upon further analysis, the paper demonstrates that Claude exhibits less position bias compared to GPT-4 when assessed on randomly-shuffled texts. Secondly, the paper reveals that Claude demonstrates a more sophisticated utilization of numbers, including basic computation, compared to other models. Additionally, it categorizes different instances of hallucination by LLMs in handling numerical data.

The paper introduces intriguing phenomenon, such as Claude exhibiting reduced position bias and superior utilizing of numeric data. However, merely reporting these observations may not contribute significantly to the scientific values. The paper would be further strengthened if it could elucidate the underlying reason for the phenomenon. Another drawback is that it has been evaluated only on a single document type -- 10-K. This raises questions about the generalizability of the findings to the other document types.

**Questions To Authors:**

- It looks like each different model received truncated texts due to the limited context window. It would be great if the paper adds the average amount of the truncated contents.

- In Section 3.1, the paper says “Specifically, we find that the GPT-4’s summary shows a high degree of condensation such as summarizing net income, net sales, and gross profit in a single sentence (Table 2)”. What is the evidence backing this claim?

- How were the tables represented? Were the HTMLs used as an input to LLM?

**Reasons To Accept:**

- The paper presents several evaluation results that may be of interest to the research community. Specifically, it demonstrates that Claude outperforms other LLMs in summarization tasks by considering actual content rather than blindly favoring beginning sentences. Moreover, Claude demonstrates a sophisticated utilization of numeric data, including the ability to perform numeric computations.

- The research on financial document summarization remains under-explored. Financial documents present a unique challenge due to their inclusion of tabular and numeric data. Therefore, evaluation results in this under-explored domain could offer valuable insights to the research community.

**Reasons To Reject:**

- The scientific contribution of the paper appears to be limited. Its primary contribution lies in contrasting the behavior of Claude2, GPT-4 and other LLMs. However, the significance of these contributions is unclear, as even acknowledged by the authors themselves, who state in the paper, "...While the position bias of GPT-4 is probably sub-optimal, it is unclear what makes a good summary, especially for such long-form inputs…"

- The generalizability of the paper's claims is also uncertain. The evaluation is based solely on one type of financial document - 10-K reports, which are publicly available and likely used during the pre-training of the LLM models. It's questionable whether the findings can be applied to other financial documents  or to any document containing tables and numbers.

- Strengthening the paper would involve providing hypotheses to explain its findings. Specifically, the paper reveals that larger models tend to utilize numbers, while smaller models do not. Additionally, it demonstrates that Claude employs numbers in a more sophisticated manner than GPT-4. Offering potential explanations for these phenomena would enhance the paper's impact and interest.

---

> ### Author Rebuttal · Authors · 2024-05-29
>
> Thank you for the insightful comments! We appreciate your recognition of financial report summarization as an under-explored domain.
>
> **Re: Scientific contribution of the paper**
>
> Our core contribution is to propose the first computational framework for characterizing summaries of multimodal long-form inputs, thus providing insights into LLM capabilities for complex real-world scenarios. While what makes a good summary remains an open question, our work is a meaningful first step forward. We will clarify that in the revision.
>
> **Re: Generalizability of the paper's claims**
>
> We acknowledge the possibility of 10-K reports used for pre-training, but there is no groundtruth summary. The challenge of summarization remains even if the model has seen these documents. We chose 10-K reports for their complexity and practical relevance.
>
> Our framework is readily transferrable to other domains. While we cannot provide formal guarantee, we expect the results from our study to be relevant and at least informative for other domains.
>
>
> **Re: Hypotheses for the findings**
>
> As the LLMs used in our work are closed, we do not know the details of the models. We propose the following hypotheses and will add to the revision.
>
> - The enhanced capabilities of larger models including fine-grained detail extraction and advanced reasoning capabilities may enable them to use numbers effectively in summaries. In contrast, smaller models might overlook complex tables and detailed numbers.
> - Claude may have more exposure to enterprise applications involving structured data and business metrics. The objectives during fine-tuning of Claude might include tasks that emphasize the importance of utilizing numerical values.
>
> **Re: Average amount of the truncated contents**
>
> We do summarization on 1,000 reports of each model, with an averaged of 436.44, 143.46, 12655.53, and 7689.75 words truncated for Claude 2.0, GPT 4, GPT 3.5, and Cohere respectively. We will update this in our final version.
>
> **Re: What is the evidence backing this claim?**
>
> This paragraph highlights our case study on abstractive sentences whose source information cannot be matched automatically. We will modify it to "Specifically, we find GPT-4 generates abstractive sentences with a high degree of condensation, such as summarizing net income, net sales, and gross profit in a single sentence (Table 2)."
>
> **Re: Representation of tables and input**
>
> Tables are represented in HTML. We use clean texts as inputs for LLMs.

---

> ### Comment · Area_Chair_dLMQ · 2024-06-03
> **[Area Chair Comment] Author Response**
>
> Dear reviewer ,
>
> Please take a look at the author's rebuttal. If the rebuttal addressed your concerns, please let the authors know about this and update your review. If not, please continue to engage with the authors and the other reviewers in the discussion forum.
>
> In particular, are your concerns about the generalizability of the approach and the paper's main contributions addressed?
>
> Other reviewers should also feel free to comment on these points!
>
> Thanks.

---

> ### Comment · Reviewer_Lb8C · 2024-06-03
>
> Thanks for the rebuttal. I was initially concerned about the generalizability of the findings and limited scientific contribution, as the paper primarily reports only behavioral differences among various LLM models. After reading the other reviews and considering the authors' feedback, I still have concerns regarding the generalizability. However, I think that the observational findings reported in the paper could be valuable to the research community. Therefore, I have updated my score.

---

> > ### Author Response · Authors · 2024-06-04
> >
> > Thank you very much for considering our feedback! We agree that generalization is a key issue and will further clarify this in Limitations.

---

### Official Review · Reviewer_cjmF · 2024-05-11

**Rating:** 6
**Confidence:** 4
**Ethics Flag:** 1

**Summary:**

This paper analyzes the use of LLMs for summarizing financial reports with a specific focus on how LLMs handle numbers since financial reports are often dense with numerical figures and a good summary needs to condense not just text but also numerical data correctly. They also do human labelled case studies on abstractive sentences and certain numerical figures found in the summary to check if these are correct or if they are hallucinations.

**Questions To Authors:**

--

**Reasons To Accept:**

1. They analyze the summaries from both an abstractive and an extractive point of view
2. They break down numerical figures into 4 types, one of those being "not found in the report". This 4th type is analyzed further to see if it was obtained by the LLM doing simple arithmetic or if it a made up hallucinated number. They report hallucinations to be minimal
3. They also compare Claude 2 and GPT 4 summaries in terms of position bias i.e. are the LLMs able to find the truly important information or are they just focussing in the beginning/end
4. They compare 4 leading LLMs including Claude 2 and GPT 4, with good conclusions and comparisons

**Reasons To Reject:**

1. While this work is a good effort in this direction, we are a long way away from ensuring that the summaries are factually correct.
2. Some more prompting methodologies such as chain of thought or speculative prompting could have been used to see if hallucinations can be fixed through LLM introspecting it's own output.
3. Most analysis is done via human labelled case studies and will be hard to reproduce

---

> ### Author Rebuttal · Authors · 2024-05-29
>
> Thank you for appreciating our work on position bias and numeric hallucinations!
>
> **Re: A long way away from ensuring factual accuracy**
>
> We agree that ensuring factual accuracy is challenging, especially when gold standards become scarce. It involves knowledge grounding, cross-verification, and incorporating human oversight, which is beyond the scope of this paper.
>
> Meanwhile, factual accuracy is not the only criterion for desirable summaries. Therefore, while our work does not directly tackle this issue, we propose a computational framework for characterizing multimodal long-form summarization. By investigating *how* LLMs approach the summarization task, our work takes a crucial step toward evaluating long-form summarization.
>
>
> **Re: Prompting**
>
> - To clarify, we find that LLMs hallucinate at a low rate when using numeric values. However, GPT-4 tends to generate long summaries with few numbers with a simple prompt. Therefore, we design experiments to enhance GPT-4's performance in extracting numeric values.
> - Following your suggestion, we designed a chain of thought (CoT) prompt and conducted experiments on Claude 2.1 and GPT-4. The experiment results on 10 reports are shown below:
>     ```
>     Model        Words	TotalNum NumDens%	A	B	C	D
>     Claude 2.1   233.70	18.90	 8.09		5.70	3.00	5.60	4.60
>     GPT-4	     323.40	11.30	 3.49		4.90	2.20	3.20	1.00
>
>     Model\Hallu%	FabNum	ArithErr   RoundErr   CtxtMis	Total
>     Claude 2.1	1.59	0.53	   0.53		2.12	4.76
>     GPT-4		0.00	0.00	   0.88		1.77	2.56
>     ```
>     The results show that while GPT-4 extracts more numbers, it still lags behind Claude 2.1. Notably, much fewer hallucinations are generated when using CoT to guide GPT-4 how to avoid specific numeric hallucinations. We will update the results in the final version of our paper.
> - We cannot find references to 'speculative prompting'. Could you please give us more information?
>
>
> **Re: Reproducibility**
>
> - Human annotation is often considered the gold standard in existing tasks. Although generating groundtruth summaries for long inputs is challenging, we use human annotation to verify whether a number is hallucinated given relevant contexts, a much more manageable task. We gave a detailed annotation protocol in Section 4.2 to facilitate reproduction, which is a common practice in human annotation.
> - Also, we did try to use GPT-4 to analyze numeric hallucinations automatically and found that GPT-4 was not able to perform this task.

---

### Official Review · Reviewer_e6Hp · 2024-05-13

**Rating:** 7
**Confidence:** 3
**Ethics Flag:** 1

**Summary:**

This paper presented a case study of long-form summarization in the finance domain in order to study LLMs’ behaviors when it comes to summarizing long input texts and numerical operations. Specifically, the authors first proposed a computational framework to characterize multimodal long-form summarization in summarizing financial reports and investigated how the information in a given financial report is utilized. Then the authors conducted a thorough inspection and evaluation of how selected LLMs handled numbers in financial reports, and proposed a taxonomy of categorizing errors related to numeric operations. The findings of the experimental results show that LLMs have strong position biases and can be mitigated by shuffling operations.

**Questions To Authors:**

1. Regarding the authors’ comment and references about LLM-generated summaries overwhelmingly preferred by humans: I think this is only applied to single document one-sentence new summarization [1]. As some previous work showed , this is definitely not the case for non-news summaries ([2]---[4]). In other words, single document summarization of news is solved != summarization is solved.

2. The authors may want to cite some prior work on chart-to-summary generation ([5]--[7]).


[1] https://direct.mit.edu/tacl/article/doi/10.1162/tacl_a_00632/119276/Benchmarking-Large-Language-Models-for-News

[2] https://dl.acm.org/doi/abs/10.1145/3477495.3531802?casa_token=g-Uw8_87GzwAAAAA:gnrbj6AxvcswhwB7Mjzumbzynm0eG6aB-A1WYPlsKhyGjn-9p12bn71uCp8rde7fxMRgHRwHTMNswg

[3] https://aclanthology.org/2023.findings-acl.593/

[4] NewSumm Keynote 1 by Kathleen McKeown

[5] https://aclanthology.org/2022.acl-long.277/

[6] https://aclanthology.org/2021.ranlp-1.183/

[7] https://link.springer.com/chapter/10.1007/978-3-319-46478-7_41

**Reasons To Accept:**

This paper is well-written and well-organized, making it very easy to follow. The problem studied in this paper is well-motivated and took on a rather challenging and urgent task in the age of LLMs. The presented study asked important questions with regard to how LLMs use information in long input text in a specific domain that contains many numbers that are important to the interpretation of the original text. I also like that the authors showed concrete examples when detailedly explaining the four types of numeric value utilization. In addition, the authors also called for new approaches or perspectives of evaluating long-form summarization when the gold standard isn’t necessarily the sole ground truth and when such ground truth is getting more and more difficult to get when the source text gets longer and longer.

**Reasons To Reject:**

I don’t have much to say here, but I have some comments and questions that I would like to bring up in the following section.

---

> ### Author Rebuttal · Authors · 2024-05-29
>
> Thank you very much for your insightful comments! We appreciate your recognition of our contribution in developing a computational framework to characterize multimodal long-form summarization as well as the challenging nature of long-form summarization!
>
> **Re: Single document summarization of news is solved != summarization is solved**
>
> We agree that claims about single document summarization based on one specific type don't necessarily generalize to other summarization tasks. As we mentioned, summarization is ubiquitous and controversial. We will further clarify this point in the revision.
>
> **Re: The authors may want to cite some prior work on chart-to-summary generation**
>
> Thank you for the pointers! We will include the relevant citations in the revision.

---

> > ### Comment · Reviewer_e6Hp · 2024-06-03
> >
> > Thanks for responding to my reviews. Best of luck with the revision.

---

### Decision · Program_Chairs · 2024-07-10

**Decision:**

Accept

**Comment:**

The paper studies the performance of LLMs on long-form summarization of financial documents. The main contributions are: (1) benchmarking LLMs on this new long document domain and characterizing their behaviors (position bias, extractiveness, etc.) (2) taxonomy of numerical errors in long-form summarization in this domain.

Pros: The error taxonomy created for numerical errors, which the paper posits are the most critical errors in this domain, is useful. The analysis of how LLMs extract and summarize information rich in numerical data is interesting.

Cons:
While the analysis of extractiveness, position bias is interesting, it is not clear what the takeaways from these observations should be. For e.g., there is no discussion of what a "good" summary is, and therefore, not clear how to interpret one LLM being XX% more extractiveness than the another. The paper can be made stronger by analyzing what *gold* summaries look like for this domain and comparing LLM performances against that standard.

The paper only focuses on one type of factual errors in the summary, and as such, it is unclear how important or prevalent these are compared to other factors. The lack of human evaluation of overall quality makes it difficult to ascertain the utility of any of these LLMs for this task.